# Engineering substrate specificity of HAD phosphatases and multienzyme systems development for the thermodynamic-driven manufacturing sugars

Chaoyu Tian [1,2,3], Jiangang Yang [1,2,3✉], Cui Liu [1,2,3], Peng Chen[1,2], Tong Zhang[1,2], Yan Men[1,2], Hongwu Ma [1,2✉], Yuanxia Sun [1,2✉] & Yanhe Ma[1,2]

Naturally, haloacid dehalogenase superfamily phosphatases have been evolved with broad substrate promiscuity; however, strong specificity to a particular substrate is required for developing thermodynamically driven routes for manufacturing sugars. How to alter the intrinsic substrate promiscuity of phosphatases and fit the "one enzyme-one substrate" model remains a challenge. Herein, we report the structure-guided engineering of a phosphatase, and successfully provide variants with tailor-made preference for three widespread phosphorylated sugars, namely, glucose 6-phosphate, fructose 6-phosphate, and mannose 6-phosphate, while simultaneously enhancement in catalytic efficiency. A 12000-fold switch from unfavorite substrate to dedicated one is generated. Molecular dynamics simulations reveal the origin of improved activity and substrate specificity. Furthermore, we develop four coordinated multienzyme systems and accomplish the conversion of inexpensive sucrose and starch to fructose and mannose in excellent yield of 94–96%. This innovative sugar-biosynthesis strategy overcomes the reaction equilibrium of isomerization and provides the promise of high-yield manufacturing of other monosaccharides and polyols.

---

[1] National Engineering Laboratory for Industrial Enzymes, Tianjin Institute of Industrial Biotechnology, Chinese Academy of Sciences, Tianjin 300308, China. [2] National Technology Innovation Center of Synthetic Biology, Tianjin 300308, China. [3]These authors contributed equally: Chaoyu Tian, Jiangang Yang, Cui Liu. ✉email: yang_jg1@tib.cas.cn; ma_hw@tib.cas.cn; sun_yx@tib.cas.cn

Haloacid dehalogenase (HAD)-like phosphatases are widespread across all domains of life and play a crucial role in the dynamical regulation of sugar-phosphate metabolite level in cell[1,2]. In nature, to recycle the essential macronutrients phosphorus necessary for sustaining central carbon metabolism[3], this distinctive superfamily has been progressively evolved with broad catalytic promiscuity and substrate ambiguity[4,5]. However, tailor-made specificity of phosphatases to a particular substrate is required in manufacturing desired chemicals[6,7]. To date, numerous enzyme engineering strategies have been developed to explore novel catalytic function to achieve the "one enzyme-multiple substrate" model. However, if the enzyme class family naturally exhibits substrate promiscuity, how to alter this intrinsic feature and meanwhile fit the classical "one enzyme-one substrate" model remains a challenge.

Biotransformation involving an isomerization reaction has been widely applied in manufacturing a series of monosaccharides, such as fructose[8], mannose[9], and rare sugars[10]. However, such monosaccharide isomerization strategy suffered from unfavorable thermodynamic equilibrium. Typically, industrial-scale manufacturing of fructose mainly relied on enzymatic isomerization of glucose with a conversion rate of less than 55%. In addition, if the isomerization products further served as substrates for producing other chemicals, then such low conversion rate would decrease the subsequent product yield. For instance, enzymatic epimerization of glucose to mannose was no more than 15%[11], which led to low mannitol yield via the chemical hydrogenation of the mixture. Therefore, it is of great interest to develop thermodynamically driven routes to manufacture sugars and bypass the reaction equilibrium of isomerization.

Phosphorylation/dephosphorylation-based cascade reaction routes provide an efficient approach to circumvent the reaction equilibrium of monosaccharide isomerization[12]. In this context, the low-cost substrate was activated to sugar-phosphate (sugar 1-phosphate or sugar 6-phosphate) by kinases or sugar phosphorylases, and then followed by the sequential conversion of such activated sugars via isomerization, epimerization, transamination, and cyclization or any other enzymatic modification reactions. In the last step, the dephosphorylation reaction catalyzed by phosphatases becomes irreversible, thereby pushing the overall reaction forward to the final product. Those routes are thermodynamically driven and in theory, the conversion rate would reach 100%[12]. As such, several in vitro synthetic enzymatic biosystems have been designed and constructed to manufacture inositol[13], fructose[14], allulose[15] and mannose[16] from inexpensive starch; higher conversion rate was obtained compared with traditional isomerization methods. However, due to the broad catalytic promiscuity of used phosphatases to phosphorylated intermediates in the biosystem, the resulting conversion rates obtained in previous studies were still far from the theoretical value. Therefore, phosphatases with strict catalytic specificity to singular target substrate are required for stoichiometrically manufacturing those monosaccharides.

In this study, we set out to engineer phosphatases for dedicated substrate specificity. Given that protein activity and substrate specificity may influence each other[17], it is still quite challenging to rapidly achieve it with excellent specificity and without activity trade-off. Here, the two catalytic properties of one phosphatase are cooperatively evolved using several structure-guided engineering approaches. Ultimately, we successfully deliver several variants that separately exhibit both enhanced preference and catalytic efficiency to three widespread C6 sugar-phosphates, namely, glucose 6-phosphate (G6P), fructose 6-phosphate (F6P), and mannose 6-phosphate (M6P). Computational simulations are performed to reveal the origin of improved activity and substrate specificity. By virtue of those beneficial mutants, we construct several thermodynamically driven synthetic systems and accomplish the near-stoichiometric conversion of inexpensive sucrose and starch to fructose and mannose with excellent yield and the highest product proportion up to now. Our work provides an efficient and promising way for manufacturing monosaccharides and polyols and facilitates overcoming the reaction equilibrium of isomerization.

## Results

**Phosphatases mining for catalytic specificity to particular substrates.** To discover phosphatases with high substrate specificity characters, general enzyme mining was performed. We at first developed a convenient phosphatase screening system comprising of four enzymes (Fig. 1a), namely, glucan phosphorylase from *Thermotoga maritima* (TmGP)[13], phosphoglucomutase from *Thermococcus kodakarensis* (TkPGM)[13], glucose 6-phosphate isomerase/mannose-6-phosphate isomerase from *Dictyoglomus thermophilum* (DtPGI/MPI)[16], and the candidate phosphatases. The first three enzymes converted maltodextrin and phosphate to a mixture of four phosphorylated sugars, namely, glucose 1-phosphate (G1P), G6P, F6P, and M6P, whereas the last phosphatases are responsible for the dephosphorylation of these sugar-phosphates. The proportion of mannose/glucose/fructose in the total monosaccharides of the dephosphorylated products ($P_{M/G/F}$) served as an index for evaluating the specificity to particular substrates. The high $P_{M/G/F}$ value represents the potential preference for corresponding substrates. This designed screening system also led to the discovery of other highly substrate-specific phosphatases by combining or altering the sugar-phosphate isomerases.

In this work, 15 thermophile-derived phosphatases were overexpressed in *Escherichia. coli*, purified by heat precipitation, and measured using the above screening system (Supplementary Fig. 1). Among these tested phosphatases, the phosphatases from *Thermotoga* sp. 38H (Pase1), *Thermoclostridium stercorarium* (Pase12) and *Petrotoga miotherma* (Pase14) presented the highest proportion for mannose ($P_M$), glucose ($P_G$), and fructose ($P_F$), respectively (Fig. 1b). The $P_M$ value of Pase1 (82%) was even higher than that of Pase5 (74.6%), as mentioned in our previous study[16]. Their catalytic activity to G1P, G6P, F6P, and M6P were measured (Supplementary Table 1). Here, to describe the specificity of phosphatase to a particular substrate in a quantitative way, we used another parameter ($S_{R1/R2}$), which was calculated by the ratio of catalytic activity toward substrate R1 over R2. A higher $S_{R1/R2}$ value represented a stronger preference for R1 relative to R2. As expected, the $S_{M6P/G6P}$ and $S_{M6P/F6P}$ values of Pase1 (also named as Ts38HM6PP) were 21 and 15, respectively (Fig. 1c), which were in accordance with its high $P_M$ value. With respect to F6P, the $S_{F6P/M6P}$ and $S_{F6P/G6P}$ values of Pase14 reached to 28 and 9, respectively, thereby confirming the preference for F6P. The generated glucose in the screening system was probably due to the dephosphorylation of G1P and G6P. In this study, the catalytic activities of Ts38HM6PP, Pase14, and Pase12 to G1P have not been detected. Therefore, the high $P_G$ value of Pase12 demonstrated its high preference to G6P, which was confirmed according to the $S_{G6P/M6P}$ and $S_{G6P/F6P}$ values. Although those three phosphatases showed tolerable substrate specificity, their measured activity was far from the required level, especially for Pase14 and Pase12 (Fig. 1d). In addition, the proportion of unwished monosaccharides in the system took up more than 15%, thereby indicating that their substrate specificity should be further improved. Based on the performance concerning enzymatic activity and substrate specificity, Ts38HM6PP was chosen for deep-seated engineering.

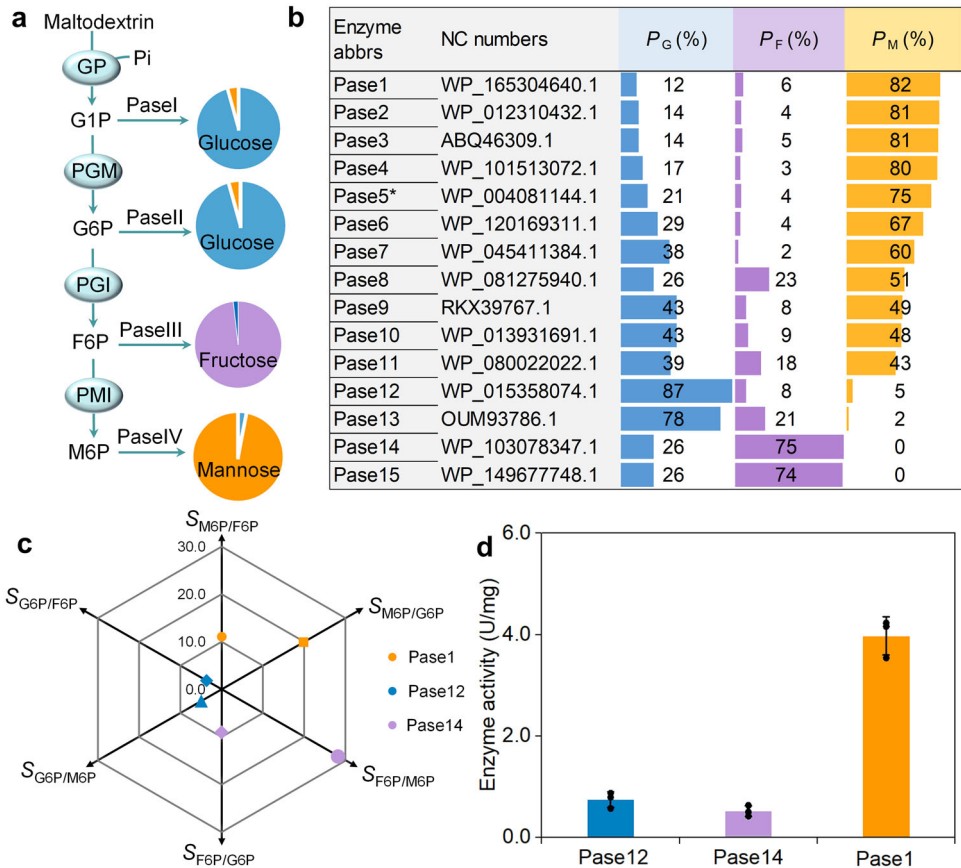

**Fig. 1 The phosphatases mining for catalytic specificity to particular substrates. a** the screening system for candidate phosphatases using maltodextrin as substrate. Pi: phosphate; Pase was the abbreviation of phosphatases. The reaction system containing 10 g/L of maltodextrin, 10 mM of PBS (pH 6.5), 5 mM of MgCl$_2$. 0.3 mg/mL of TmGP, 0.2 mg/mL of TkPGM, 0.2 mg/mL of DtPGI/MpI, and 60 μL of heat-treated or purified Pases was conducted at 55 °C for 4 h. **b** The $P_{G/F/M}$ values for different phosphatases were presented. For example, the $P_M$ value was calculated by the ratio of mannose concentration to total monosaccharides concentration. $P_G$ value indicates the proportion of glucose, and the $P_F$ value presents the proportion of fructose. **c** The $S_{R1/R2}$ values for Pase1, Pase12 and Pase14. The $S_{M6P/F6P}$ of Pase1 displayed the ratio of activity of Pase1 to M6P over F6P. **d** The enzyme activity of Pase1 to M6P, Pase12 to G6P and Pase14 to F6P were presented. Data are presented as mean values +/−SD ($n = 3$ independent experiments). Source data for **d** are provided as a Source Data file.

**Structure-guided engineering to enhance or create preference to dedicated substrate**. Although HAD-like phosphatases share very low sequence similarity, its members possess a highly conserved core domain, such as Rossmann-like fold and metal-binding motif, which is required for enzymatic catalysis; and the variable cap domain is responsible for the substrate recognition[18,19]. The Ts38HM6PP possessed 89.14% sequence similarity with Pase5 from *T. maritima*, whose crystal structure has been released in the database (PDB 1NF2)[20]. A homology model for Ts38HM6PP was built with 1NF2 as template. After binding with M6P, an "open and closed" conformation change was observed for cap domain of Ts38HM6PP (Fig. 2a). The cap-open conformation allows the active site access to the solvent, thus facilitating substrate binding and product release; in the cap-closed conformation, nucleophilic attack for catalysis occurs[21]. The residues related to this conformational flexibility may affect substrate recognition and binding[22]. Here, we examined the B-factor values for both open and closed conformations and discovered that residues 124–127 in the cap domain presented high B-factor values in the cap-open conformation but low values in cap-closed conformation upon substrate binding (Fig. 2a). Among these four sites, S126 directly interacted with the mannose cycle of M6P, demonstrating its potential role in substrate recognition. To obtain mutants for recognizing particular substrates, saturation mutagenesis was performed at the S126 site, and mutants were measured using the

screening system (Fig. 2d and Supplementary Fig. 2). Impressively, the mutations of S126W, S126E, and S126D significantly increased $P_G$ value, and the substitution of Ser with Phe (S126F), Tyr (S126Y), and His (S126H) displayed the improved $P_F$ value. The Ser and Thr located at this site maintained the high $P_M$ value of the Ts38HM6PP (WT). Among those beneficial mutations, mutation S126D increased the glucose proportion ($P_G$) to 76%, altered the preference from M6P of WT to G6P, and improved its specificity to G6P over F6P according to the $S_{G6P/M6P}$ and $S_{G6P/F6P}$ data (Figs. 3a–c). Particularly, the single-site mutation S126F increased the $P_F$ value from 6% to 95% and generated a 350-fold preference switch from M6P of WT to F6P according to the $S_{F6P/M6P}$ value of 17 (Fig. 3b). Moreover, although the WT showed similar specificity to F6P and G6P, mutant S126F altered this feature and increased the $S_{F6P/G6P}$ value to 43 (Fig. 3c), thereby, demonstrating enhanced preference for F6P over G6P. This mutation also improved the catalytic efficiency to F6P by 4.3-fold relative to WT without activity trade-off, and almost lost its catalytic ability to G6P simultaneously (Table 1). This finding suggested that the conformational flexibility region related to substrate binding can serve as hotspot position(s) for substrate specificity engineering.

The cap domain contains the specificity determinants for particular substrates[21]; however, how to locate their precise position and generate the desired mutations is still a difficult task. Here, in silico saturation mutagenesis was performed for the

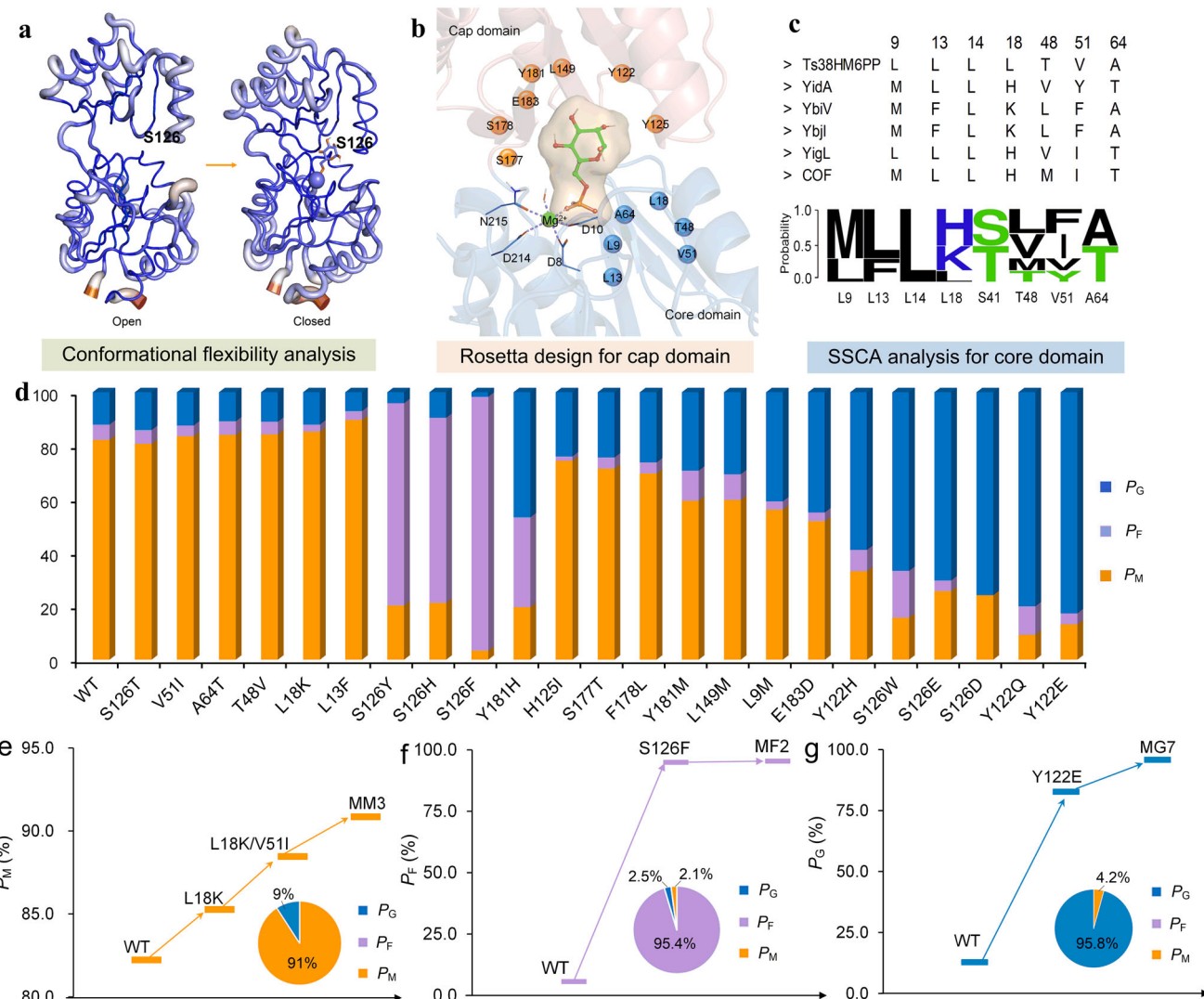

**Fig. 2 Enzyme engineering of phosphatases for dedicated specificity. a** B-factor values change for both open and closed conformations. Residues 126 has crudeness sticks indicate high B-factor values in the cap-open conformation, but fineness sticks represent low values in cap-closed conformation upon substrate binding. **b** The mutation sites in cap domain. **c** Specificity-based sequence conservative analysis strategy for substrate specificity engineering in core domain. The protein sequences and accession codes for phosphatases YigL, COF, YbiV, Ybjl, and YidA were presented in Supplementary Data 1. **d** The $P_{G/F/M}$ values for different single-site mutants. **e–g** presented improved $P_M$, $P_F$, and $P_G$ values after combinational mutations, respectively. MM3: L18K/V51I/T48V; MF2: S126F/Y181H; MG7: Y122E/L9M/E183D/H125I/F178L/S177T/Y181M. Source data for **d–g** are provided as a Source Data file.

residues, which interacted with the sugar ring of phosphorylated sugars, to test their binding free energy for each substrate of M6P, F6P, and G6P. Twenty candidate mutants at seven sites (Y122, H125, L149, S177, F178, Y181, and E183) were selected due to their lower binding free energy than the WT (Fig. 2b and Supplementary Table 2). Their proportion $P_{M/G/F}$ values were experimentally measured in the screening system (Fig. 2d and Supplementary Fig. 2). Many mutations increased the glucose content in all monosaccharides mixture, especially for the mutation of Y122E, which presented the highest $P_G$ value (82.7%) (Fig. 2d). The mutation Y181H showed improved $P_F$ value compared with WT, thereby indicating the beneficial effect for F6P.

The core domain of phosphatases contains several conserved motifs that are responsible for binding the substrate phosphoryl group and coordinating magnesium ion for nucleophilic attack[23]. The residues around the active site are variable and may influence catalytic performance, such as activity and substrate specificity. To date, several phosphatases have been characterized with

different preference to M6P, G6P, or F6P[2]. Here, sequence alignment of those phosphatases with Ts38HM6PP was performed to analyse the relationship between sequence conservation and activity or specificity (Supplementary Fig. 3). Seven residues (L9, L13, L18, S41, T48, V51, and A64) of Ts38HM6PP were selected and mutated to other amino acids occurring at the corresponding site (Fig. 2b, c). The $P_{M/G/F}$ results for those mutants showed that five mutations (V51I, A64T, T48V, L18K, and L13F) presented increased $P_M$ values (Fig. 2d). Although the $P_M$ value of mutation L13F was the highest, this mutation presented low mannose production (Supplementary Fig. 4). Compared with the WT, mutation L18K not only increased the catalytic efficiency ($k_{cat}/K_m$) by 3-fold but also improved its preference for M6P according to the enhanced $S_{M6P/F6P}$ and $S_{M6P/G6P}$ values (Table 1, Fig. 3a, b). Interestingly, the substitution of Leu with Met at site 9, which is close to catalytic sites D8 and D10, contributed to the increase in the preference for G6P compared with the WT (Fig. 2d). At this point, this specificity-based sequence conservative analysis (SSCA) strategy provides a

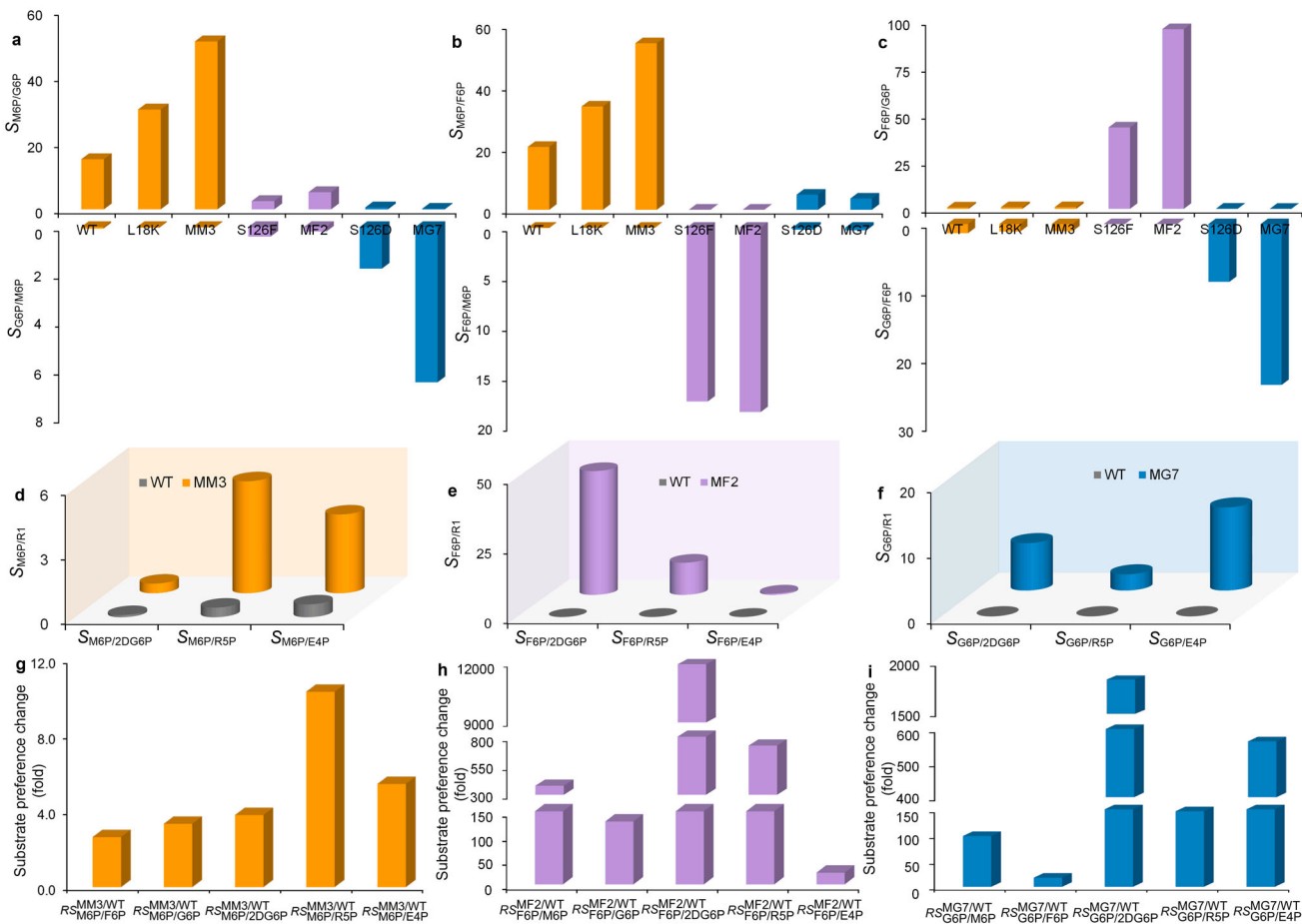

**Fig. 3 The SR1/R2 and $RS_{R1/R2}^{Mutant/WT}$ values of the WT and mutants. a–c** represented the $S_{M6P/G6P}$ and $S_{G6P/M6P}$ values, $S_{M6P/F6P}$ and $S_{F6P/M6P}$ values, and $S_{G6P/F6P}$ and $S_{F6P/G6P}$ of WT and different mutants, respectively. **d** The $S_{M6P/R2}$ values of WT and mutant MM3 for M6P over substrates 2DG6P, R5P and E4P. **e** The $S_{F6P/R2}$ values of WT and mutant MF2 to F6P over other three substrates. **f** The $S_{G6P/R2}$ values of WT and mutant MG7 to G6P over other three substrates. **g–i** represented the substrate preference change compared with WT. For example, $RS_{F6P/M6P}^{MF2/WT}$ value was calculated by the ratio of $S_{F6P/M6P}$ of MF2 to that of the WT. 2DG6P:2-deoxy-D-glucose-6-phosphate; R5P: D-ribose-5-phosphate; E4P: D-erythrose-4-phosphate. Source data for Fig. 3 are provided as a Source Data file.

feasible way to improve catalytic activity and alter substrate preference through engineering the amino acids around active center. In summary, by enzyme engineering of amino acids in the cap and core domains, we gained 13 single-site mutations with increased preference for G6P, 4 single-site mutations with improved preference for F6P, and 5 single-site mutations with enhanced preference for M6P.

**Enhancing specificity and activity to particular substrate by combinational mutation.** To further increase the specificity and catalytic efficiency to particular substrates, we assembled the beneficial single-site mutations. For M6P, mutation L18K served as a template to combine V51I, A64T, and T48V, respectively. Three generated double mutants displayed improved $P_M$ values compared with L18K, especially for L18K/V51I; the $P_M$ value reached to 88.5% (Supplementary Fig. 5). Mutations T48V and A64T were then performed with L18K/V51I as a starting point, and the generated three mutants L18K/V51I/T48V (MM3) increased the $P_M$ value to 90.9% (Fig. 2e). In this study, to describe the substrate specificity switch or enhancement compared with the WT in a quantitative way, we introduced a parameter $RS_{R1/R2}^{Mutant/WT}$, which was calculated by the ratio of $S_{R1/R2}$ value of mutant over WT. A higher $RS_{R1/R2}^{Mutant/WT}$ value indicated larger improvement for this mutant relative to WT with the

respect to substrate preference for R1 over R2. The $RS_{M6P/G6P}^{MM3/WT}$ and $RS_{M6P/F6P}^{MM3/WT}$ values of mutant MM3 were 3.3 and 2.7, respectively, suggesting the enhanced preference for M6P of mutant MM3 (Fig. 3a, b, g). In addition, this mutant displayed 11.5-fold increase in catalytic efficiency ($k_{cat}/K_m$) and 2-fold increase in substrate affinity ($K_m$) to M6P compared with WT (Table 1 and Supplementary Fig. 6).

With respect to F6P, mutation Y181H was carried out using F6P-specific S126F as template. The resulting mutant S126F/Y181H (MF2) further improved the preference for F6P relative to S126F according to the $S_{F6P/M6P}$ and $S_{F6P/G6P}$ values (Fig. 3b, c). In the thermodynamic-driven biosystem for biosynthesis of fructose from starch and sucrose, F6P was generated from G6P; therefore, phosphatases with strict preference for F6P versus G6P were required. In this study, our engineered mutant MF2 displayed a high $P_F$ value of 95.4% (Fig. 2f) and very low activity to G6P (Supplementary Table 3), thereby indicating that it was a fantastic candidate for application. The enhanced preference for F6P over M6P and G6P was due to the fact that the catalytic efficiency to F6P increased by 11.8-fold; meanwhile, that to G6P and M6P decreased by 14.6- and 41.2-fold, respectively, compared with the WT (Table 1 and Supplementary Fig. 7). The mutant MF2 displayed the $RS_{F6P/M6P}^{MF2/WT}$ and $RS_{F6P/G6P}^{MF2/WT}$ of 378 and 129, respectively (Fig. 3h), indicating great achievement in

**Table 1 The kinetic parameters for Ts38HM6PP and its variants with M6P, F6P, and G6P.**

| Substrates | Variants | $K_m$ (mM) | $k_{cat}$ (s$^{-1}$) | $k_{cat}/K_m$ (M$^{-1}$·s$^{-1}$) |
|---|---|---|---|---|
| M6P | WT | 10.5 ± 1.2 | 4.3 ± 0.2 | 409.6 ± 25.0 |
| | L18K | 7.2 ± 0.9 | 8.9 ± 1.2 | 1228.8 ± 18.1 |
| | MM3 | 5.0 ± 0.4 | 23.5 ± 2.1 | 4686.4 ± 54.3 |
| | S126F | 9.8 ± 0.6 | 0.06 ± 0.01 | 6.3 ± 0.5 |
| | MF2 | 13.3 ± 1.2 | 0.13 ± 0.02 | 9.9 ± 0.3 |
| | S126D | 13.0 ± 2.1 | 0.30 ± 0.04 | 23.1 ± 0.5 |
| | MG7 | 17.9 ± 1.3 | 0.35 ± 0.03 | 19.2 ± 0.1 |
| F6P | WT | 13.8 ± 1.6 | 0.2 ± 0.05 | 15.2 ± 1.0 |
| | L18K | 11.8 ± 1.1 | 0.3 ± 0.04 | 22.3 ± 4.0 |
| | MM3 | 6.9 ± 0.8 | 0.4 ± 0.04 | 62.7 ± 5.5 |
| | S126F | 16.7 ± 1.2 | 1.1 ± 0.1 | 65.9 ± 2.9 |
| | MF2 | 15.1 ± 1.4 | 2.7 ± 0.1 | 179.0 ± 6.0 |
| | S126D | 19.3 ± 2.1 | 0.07 ± 0.02 | 3.6 ± 0.8 |
| | MG7 | 17.0 ± 3.5 | 0.08 ± 0.03 | 4.6 ± 0.7 |
| G6P | WT | 9.3 ± 2.2 | 0.3 ± 0.05 | 29.2 ± 3.0 |
| | L18K | 11.0 ± 0.9 | 0.3 ± 0.05 | 26.6 ± 2.8 |
| | MM3 | 4.8 ± 0.6 | 0.5 ± 0.04 | 97.1 ± 2.9 |
| | S126F | 27.4 ± 1.2 | 0.02 ± 0.01 | 0.7 ± 0.1 |
| | MF2 | 15.1 ± 0.8 | 0.03 ± 0.01 | 2.0 ± 0.3 |
| | S126D | 12.2 ± 0.9 | 0.56 ± 0.1 | 46.3 ± 7.2 |
| | MG7 | 17.4 ± 1.3 | 2.8 ± 0.2 | 159.0 ± 1.7 |

preference switch from M6P or G6P of the WT to F6P of MF2 have been made by substrate specificity engineering.

To obtain combinational mutants with enhanced preference to G6P, we first assembled two single-site G6P-specific mutants Y122E and S126D together. However, this combination decreased the proportion of glucose in the screening system (Supplementary Fig. 5). We then sequentially combined the abovementioned G6P-specific single-site mutants L9M, E183D, H125I, F178L, S177T, Y181M, and L149M with Y122E as the starting point. Desirably, mutant Y122E/L9M/E183D/H125I/F178L/S177T/Y181M (MG7) increased the $P_G$ value to 95.8% (Fig. 2g). According to the $S_{G6P/M6P}$ and $RS_{G6P/M6P}^{MF2/WT}$ values of mutant MG7, a 93-fold switch in substrate specificity from M6P to G6P was generated (Fig. 3a–i). In particular, its $S_{G6P/F6P}$ value reached 24, thereby, indicating enhanced preference for F6P over G6P (Fig. 3c). This G6P-specific mutant also increased the catalytic efficiency to G6P by 5.4-fold compared with the WT while displaying low enzyme activity to F6P and M6P (Table 1, Supplementary Table 3 and Supplementary Fig. 8).

We successfully obtained three mutants, namely, MM3, MF2, and MG7 with strong specificity to the particular substrates of M6P, F6P, and G6P, respectively (Supplementary Fig. 9). To test whether these mutants still held the strong specificity to the desired substrate versus other phosphorylated sugars, their catalytic activities to four typical substrates with different carbon numbers, including N-acetyl-D-glucosamine-6-phosphate (AG6P), 2-deoxy-D-glucose-6-phosphate (2DG6P), D-ribose-5-phosphate (R5P), and D-erythrose-4-phosphate (E4P), were measured. The specificity indexes $S_{R1/R2}$ were correspondingly calculated. As expected, mutants MM3, MF2, and MG7 all displayed enhanced preference versus the measured substrates compared with WT according to the $S_{R1/R2}$ values (Fig. 3d–f). To our surprise, the activity of WT to 2DG6P was 9-fold higher than that to M6P indicating its stronger preference for 2DG6P than M6P (Supplementary Table 3 and Supplementary Fig. 10). Moreover, the enzyme activity of MM3 to 2DG6P reached 55.2 U/mg, which presented the highest value, thereby suggesting its potential in manufacturing 2-deoxy-D-glucose. Whereas, mutant MF2 and MG7 presented very low activity to 2DG6P. According

to the $RS_{F6P/2DG6P}^{MF2/WT}$ values of MF2 and $RS_{G6P/2DG6P}^{MG7/WT}$ values of MG7, a nearly 12000-fold switch from preference to 2DG6P of WT to F6P of MF2 and an 1800-fold switch to G6P of MG7 were observed (Fig. 3h, i, and Supplementary Table 3).

**Shedding light on the origin of enhanced specificity and activity.** To elucidate the evolution of enhanced activity and extensive switch in specificity, molecular dynamic (MD) simulations were performed to reveal the representative binding conformation in enzyme-substrate complex, and the substrate-binding free energy was calculated using molecular mechanics generalized born surface area (MM/GBSA) method. From the representative structure of M6P, it can be seen that the sugar ring of M6P formed three hydrogen bonds with D10, S126, and S177 in WT (Fig. 4a). Mutation L18K enables its side chain stretch into the substrate-binding pocket and contributed to the generation of the fourth H-bond with the C2-OH of the sugar ring (Fig. 4b). Additional introduction of more bulky hydrophobic mutations T48V and V51I around the active site in M3, although not directly involved in substrate binding, further decreased the binding energy of MM3 to M6P (−57.37 kcal/mol) compared with that of mutant L18K (−51.05 kcal/mol) (Fig. 4c). This finding suggested that the enhanced preference and catalytic efficiency of MM3 to M6P can be derived from the stronger H-bond interaction between the sugar ring of M6P and MM3 compared with WT. With respect to F6P, replacing the Ser of 126 with Phe introduced the π-π interaction with Y122 and pushed Y122 in an orientation to generate two new strong hydrogen bonds with C1-OH and C2-OH of F6P (Figs. 4d, e). Mutation Y181H in MF2 contributed to form another H-bond with the C2-OH of F6P, correspondingly increasing its binding affinity to F6P compared with single-site mutant S126F (Fig. 4f). Those additional H-bond interactions between F6P and H181 and Y122 are responsible for the enhanced catalytic efficiency and preference switch from M6P of WT to F6P of S126F and MF2. Likewise, the significant preference switch to G6P may be due to the five H-bond interactions between G6P and MG7; while only two interactions were detected for the WT (Fig. 4g–i). According to the representative binding conformation, the H-bond formation at a particular locus, such as 18, 126, 122, 183, and 181, may influence substrate specificity to the tested substrates M6P, G6P, and F6P. In summary, our results suggested that engineering the amino acid around the sugar ring by forming strong H-bond interactions would provide an effective approach to enhance or alter the substrate specificity of phosphatases.

**Thermodynamic-driven manufacturing fructose from sucrose and starch.** Industrial manufacturing of fructose depended primarily on the isomerization of glucose with the conversion of 42–55% in a single reaction process[8]. To address this challenge, a thermodynamically driven biosystem based on phosphorylation and dephosphorylation cascade reactions was developed to produce fructose from low-cost starch[14,24]. However, due to the nonspecific catalytic ability of phosphatases to F6P and G6P, the product yield was only 43%. Here, based on the engineered mutant MF2, we reconstructed this in vitro synthetic biosystem (Fig. 5a), which contained four core enzymes, as follows: i) TmGP, which catalyzes the phosphorolysis of amylopectin to give G1P in the presence of inorganic phosphate; ii) TkPGM, which converts G1P to G6P; iii) glucose 6-phosphate isomerase from *Thermus thermophilus* (TtPGI), which converts G6P to F6P[25]; and iv) F6P-specific phosphatase MF2, which catalyzes the exclusive dephosphorylation of F6P to fructose (Supplementary

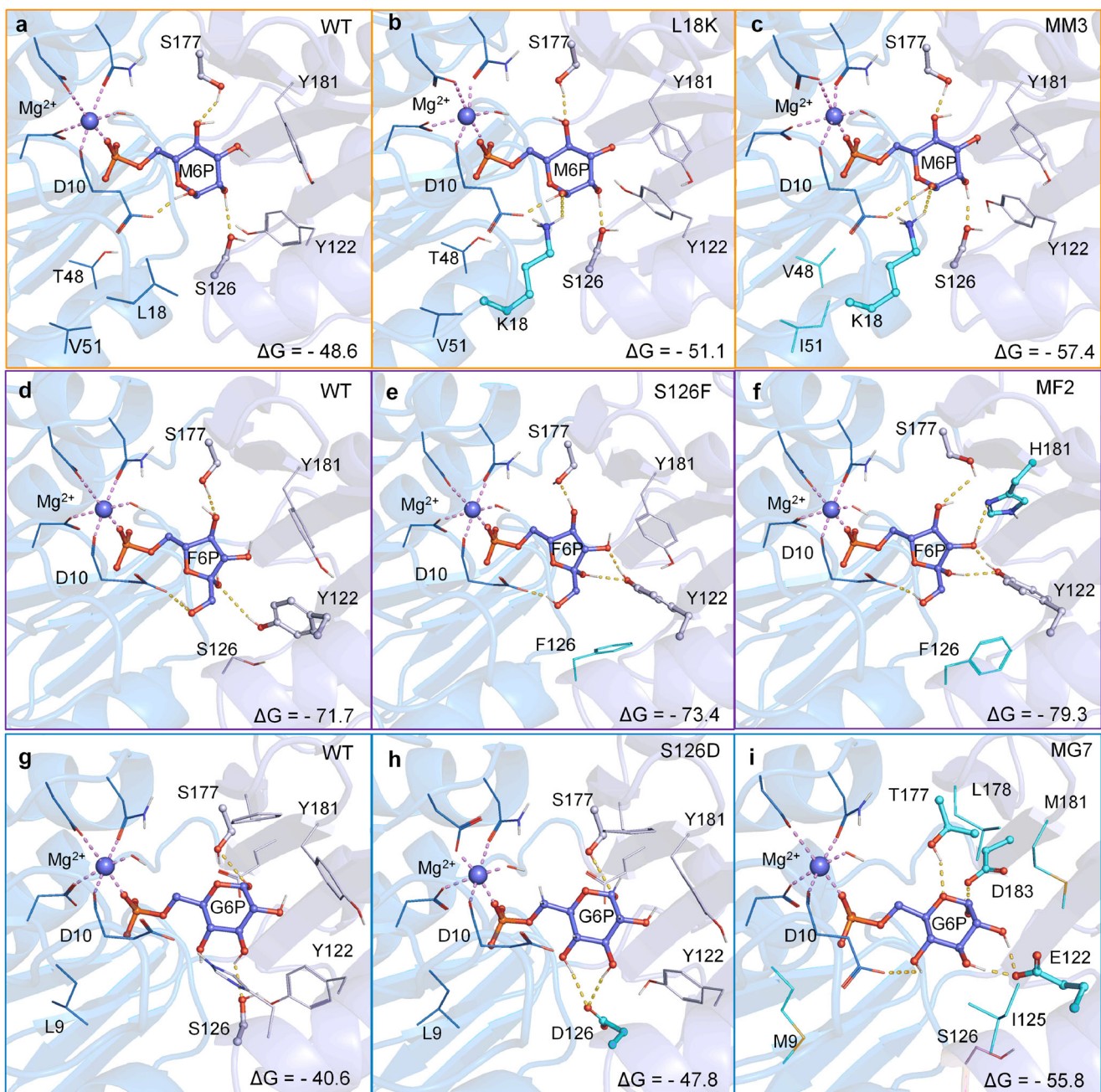

**Fig. 4 The representative binding conformation and binding energy (ΔG°) of the WT and mutants to M6P, F6P, and G6P. a–c** represented that of WT, L18K and MM3 for M6P, respectively; **d–f** represented that of WT, S126F and MF2 for F6P, respectively; **g–i** represented that of WT, S126D and MG7 for G6P, respectively.

Table 4 and Supplementary Fig. 11). The released inorganic phosphate was reutilized by TmGP, which contributed to the generation of a reaction circle for manufacturing fructose. In addition, to achieve the stoichiometric conversion of maltodextrin to fructose, we also used three auxiliary enzymes, as follows: isoamylase from *Sulfolobus tokodaii* (StIA), which debranches α−1,6-glucosidic linkages of maltodextrin to amylopectin; 4-α-glucanotransferase from *Thermococcus litoralis* (Tl4GT) which transfers the maltose/maltotriose to longer maltodextrin[13], that can be used by TmGP for G1P generation; and polyphosphate glucokinase from *Thermobifida fusca* (TfPPGK), which catalyzes the phosphorylation of glucose to G6P[26], that would be consumed by the reaction cycle. The overall Gibbs energy change (ΔG°) of this synthetic module was −30.5 kJ/mol under the standard conditions (Fig. 5a), indicating that this route was thermodynamically favorable.

The proof-of-concept experiment of this biosystem using StIA, TmGP, TkPGM, TtPGI, and MF2 gave 7.6 g/L (42.2 mM) of fructose from 10 g/L maltodextrin (55.5 mM glucose equivalent), corresponding to a product yield of 76% and a high $P_F$ value of 96% (Table 2). Employment of Tl4GT and TfPPGK consumed the residual maltose and maltotriose and contributed to the improvement of the conversion yield to 95%, which was the highest value obtained by a single biotransformation process (Table 2). When a high concentration of maltodextrin (100 g/L) was used in the reaction medium, this system produced 92.9 g/L fructose (516 mM) with an overall yield of 93% after reaction for 24 h and generated an excellent $P_F$ value of 98.8% (Fig. 5c).

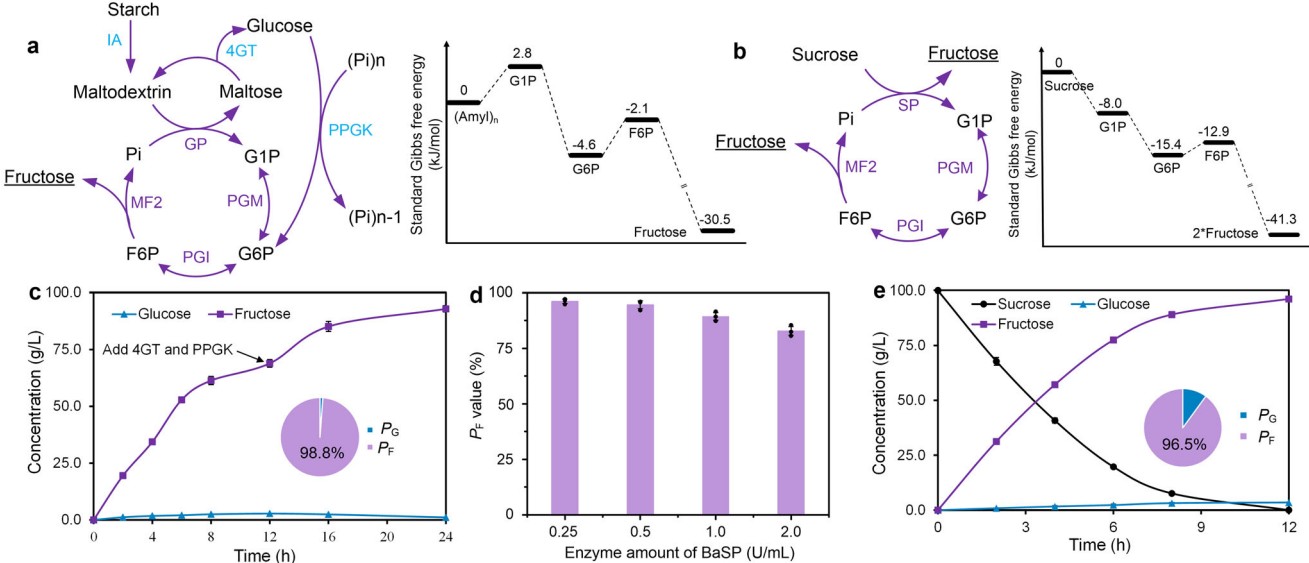

**Fig. 5 Biosynthesis of fructose from starch and sucrose by constructing multienzyme systems. a** Thermodynamically-driven manufacturing fructose from starch. The standard Gibbs free energy change of each and overall reaction was presented. The ΔG° value was freely available on the website. http://equilibrator.weizmann.ac.il. (Pi)n: hexametaphosphate, (Amyl)n: maltodextrin. **b** Thermodynamically-driven manufacturing fructose from sucrose. **c** Biosynthesis of fructose under 100 g/L of maltodextrin. The reaction medium containing maltodextrin (100 g/L), PBS buffer (30 mM, pH 6.5), MgCl$_2$ (5 mM), TmGP (10 U/mL), TkPGM (5 U/mL), TtPGI (10 U/mL), and MF2 (10 U/mL) was carried out at 55°C for 24 h. After reaction for 12 h, 5.0 U/mL of Tl4GT, 5.0 U/mL of TfPPGK, and 50 mM of hexametaphosphate were added to the reaction solution. **d** The effect of enzyme amount of BaSP on $P_F$ values. **e** Biosynthesis of fructose under 100 g/L of sucrose. The reaction medium comprising of 100 g/L of sucrose, 20 mM of PBS buffer (pH 6.5), 5 mM of MgCl$_2$, 2.5 U/mL of BaSP, 5 U/mL of TkPGM, 10 U/mL of TtPGI, and 10 U/mL of MF2 was carried out at 55 °C for 12 h. Data are presented as mean values +/−SD ($n = 3$ independent experiments). Source data for **c–e** are provided as a Source Data file.

**Table 2 Comparison of isomerization methods and thermodynamically driven routes for fructose and mannose production.**

| | Substrates | Catalysts | Products | Conversions (%) | References |
|---|---|---|---|---|---|
| Isomerization routes | Sucrose | Sucrase | Fructose | 50 | 27 |
| | Glucose | GI | Fructose | 42–55 | 8 |
| | Glucose | (NH$_4$)$_2$MoO$_4$ | Mannose | 30–40 | 29 |
| Thermodynamic-driven routes | Maltodextrin | StIA, TmGP, TkPGM, TtPGI, MF2 | Fructose | 76[a] | This study |
| | Maltodextrin | StIA, TmGP, TkPGM, TtPGI, MF2, Tl4GT, TfPPGK | Fructose | 95[a] | This study |
| | Sucrose | BaSP, TkPGM, TtPGI, MF2 | Fructose | 96[a] | This study |
| | Maltodextrin | StIA, TmGP, TkPGM, DtPGI/PMI, TmM6PP, Tl4GT, TfPPGK | Mannose | 81[a] | 16 |
| | Maltodextrin | StIA, TmGP, TkPGM, DtPGI/PMI, MM3 | Mannose | 74[a] | This study |
| | Maltodextrin | StIA, TmGP, TkPGM, DtPGI/PMI, MM3, Tl4GT, TfPPGK | Mannose | 94[a] | This study |
| | Sucrose | BaSP, TkPGM, DtPGI/PMI, MM3 | Mannose | 45[a] | This study |

[a]The results were obtained under the substrate concentration of 10 g/L.

Enzymatic hydrolysis of widespread sucrose presented another primary route for industrial production of fructose, especially in a sugarcane or a beet growing area; however, such process still suffered from low yield of 50% in a single reaction process[27]. The disaccharide sucrose is comprised of one unit of glucose and fructose. If the glucose unit in sucrose could be converted to fructose, the yield of fructose formation would reach to 100% in theory. Therefore, inspired by the beneficial effect of MF2 in manufacturing fructose from starch, we reconstructed another synthetic biosystem by replacing the TmGP with sucrose phosphorylase from *Bifidobacterium adolescentis* (BaSP)[28] to produce fructose from sucrose (Supplementary Fig. 11). By comparing with the biosystem for starch conversion, this sucrose conversion system is found to be more thermodynamically favorable according to the overall Gibbs energy change (ΔG°). Only four core enzymes were employed without necessary for other auxiliary enzymes (Fig. 5b). The proof-of-concept

experiment for this cascade reaction produced 8.8 g/L of fructose and 1.2 g/L of glucose from 10 g/L sucrose with a $P_F$ of 88%, which was lower than the desired value. It was resulted from that the used BaSP exhibited dephosphorylation activity to G1P (Supplementary Fig. 12). Then, the amounts of BaSP in the reaction system were optimized (Fig. 5d), and a higher $P_F$ of 96.5% for fructose production (10.1 g/L) was obtained when half the amount of BaSP was used (Table 2). To investigate the potential of this biosystem for industrial fructose production, the reaction process was conducted with a high sucrose concentration of 100 g/L. After reaction for 24 h, the system produced 96.2 g/L of fructose with product yield of 92% (Fig. 5e). Here, we noted that the conversion yields under high substrate concentration of 100 g/L maltodextrin and sucrose were lower than that under low substrate concentration, which was probably due to the Maillard reaction between sugars and enzymes under the reaction medium[29,30].

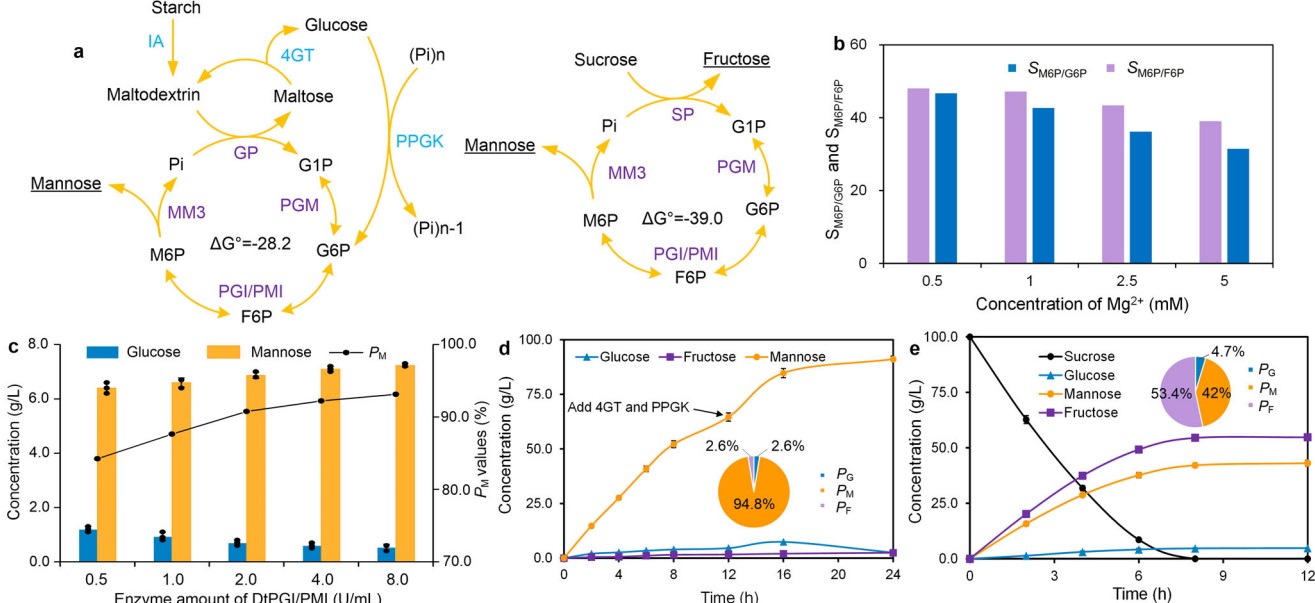

**Fig. 6 Biosynthesis of mannose from starch and sucrose in multienzyme cascade systems. a** Thermodynamically-driven manufacturing mannose from starch. **b** The effect of $Mg^{2+}$ concentration on $S_{M6P/F6P}$ and $S_{M6P/G6P}$ values of mutant MM3. **c** The effect of enzyme amount of DtPGI/PMI on mannose production and $P_M$ values. **d** Biosynthesis of mannose under 100 g/L of maltodextrin. The reaction medium containing 100 g/L of maltodextrin, 30 mM of PBS buffer, 1 mM of $MgCl_2$, 10 U/mL of TmGP, 5 U/mL of TkPGM, 20 U/mL of DtPGI/MPI, and 10 U/mL of MM3 was carried out at 55 °C for 24 h. After reaction for 12 h, 5.0 U/mL of Tl4GT, 5.0 U/mL of TfPPGK, and 50 mM of hexametaphosphate were added to the reaction solution. **e** Biosynthesis of mannose under 100 g/L of sucrose. The reaction medium comprising of 100 g/L of sucrose, 20 mM of PBS buffer, 1 mM of $MgCl_2$, 2.5 U/mL of BaSP, 5 U/mL of TkPGM, 20 U/mL of DtPGI/MPI, and 10 U/mL of MM3 was carried out at 55°C for 12 h. Data are presented as mean values $+/-$ SD ($n = 3$ independent experiments). Source data for **b–e** are provided as a Source Data file.

**High-yield production of mannose.** Nowadays, industrial large-scale synthesis of mannose still depends on the epimerization of glucose, which suffered from low yield 30-40% in a single reaction process[31]. We have presented the thermodynamically driven synthetic system for manufacturing mannose from starch using phosphatase Pase5 (Supplementary Fig. 13a); however, the $P_M$ value was only 74.6% and the conversion yield was 81%[16]. To achieve the high-yield production of mannose, we constructed another synthetic biosystem by replacing Pase5 with high M6P-specific mutant MM3 (Fig. 6a). Likewise, the system comprised four core enzymes, namely, TmGP, TkPGM, glucose 6-phosphate isomerase/mannose-6-phosphate isomerase from *Dictyoglomus thermophilum* (DtPGI/MPI), and M6P-specific phosphatase MM3 (Supplementary Fig. 11). As expected, this biosystem produced 6.8 g/L mannose (37.8 mM) from 10 g/L maltodextrin with a conversion rate of 68.7% and $P_M$ value of 90%. It produced 0.7 g/L of (3.9 mM) glucose, and meanwhile, fructose was at an undetectable level. We discovered low $Mg^{2+}$ concentration increased the substrate specificity of MM3 to M6P according to the $S_{M6P/F6P}$ and $S_{M6P/G6P}$ values (Fig. 6b). Therefore, the $Mg^{2+}$ concentration in the medium was adjusted to 1 mM; this contributed to the increase of the $P_M$ of mannose to 94%. Considering that the conversion of G6P to M6P was thermodynamically unfavorable in this cascade reaction routes, therefore, the usage of DtPGI/PMI was optimized (Fig. 6c). The results showed that using the double amount of DtPGI/PMI (4 U/mL) increased the mannose yield to 74% (Table 2). Additional usage of other auxiliary enzymes Tl4GT and TfPPGK improved the conversion rate to 94%, which was higher than that in the previous study[16]. We then performed the cascade reaction at a high concentration of maltodextrin (100 g/L) and obtained 91.1 g/L of mannose with a conversion of 91% after reaction for 24 h. The fructose production maintained low yield (Fig. 6d).

In parallel, to manufacture mannose from sucrose, we designed another biosystem in which four enzymes BaSP, TkPGM, DtPGI/MPI, and MM3 were employed (Fig. 6a and Supplementary Fig. 13b). The proof-of-concept experiment for the biosystem was conducted with 10 g/L sucrose as substrate and produced 4.7 g/L of mannose, 5.3 g/L of fructose, and 0.5 g/L of glucose. The conversion of sucrose to mannose was 45%, representing the 90% value of the theoretical value (Table 2). When a high concentration of sucrose (100 g/L) was used, the biosystem produced 43.1 g/L of mannose and presented a 42:53.4:4.7 ratio of mannose:fructose:glucose (Fig. 6e).

## Discussion

In nature, HAD superfamily phosphatases have been evolved with broad catalytic promiscuity to various phosphorylated substrates[32]. This property has become the bottleneck for thermodynamically driven manufacturing monosaccharides based on phosphorylation/dephosphorylation cascade reaction routes, which have been approved as superable approaches to circumvent the thermodynamic limitation of isomerization. How to alter this intrinsic substrate ambiguity and concentrate its catalytic ability on a particular substrate are not a trivial task. Although the efforts have been made to increase the substrate preference of one HAD phosphatase; however, significant preference switch has not been achieved[33]. To address these challenges, we successfully narrowed the substrate scope of one phosphatase by virtue of an efficient substrate preference screening system and structure-guided engineering approach. We presented several variants with dedicated specificity to the three most widespread phosphorylated sugars, namely, G6P, F6P, and M6P, while simultaneously with enhancement in catalytic efficiency. The substrate specificity engineering strategy, which is based on "open-closed" conformational flexibility, binding free energy analysis for amino acid

around sugar ring in cap domain, and sequence-based SSCA approach for core domain, provides a promising way to enhance or alter the substrate specificity of other phosphatases for strict and tailor-made substrate specificity. This rational design strategy was similar to the recently published Focused Rational Iterative Site-specific Mutagenesis (FRISM) method, which has been applied to engineer a lipase to access all possible stereoisomers of chiral esters[34,35]. MD analysis demonstrated that the origin of the enhanced or extensive switch in specificity was strong H-bond formation between the sugar ring of phosphorylated sugars and enzymes. More precise rational design through introduction of specific H-bond interactions with the sugar ring and enhancement of hydrophobic interactions at distal amino acid would contribute to the further increase in the substrate specificity to a particular substrate.

Fructose, a monosaccharide found naturally in fruits and some vegetables, exhibits excellent physical and functional properties, such as high and unique sweetness intensity, low glycemic index, Maillard browning, and flavor development, compared with another table sugar sucrose[36,37]. It has been extensively applied to the food and pharmaceutical industries. Typically, high fructose corn syrup (HFCS) is widely used as sucrose substitute in soft drinks[38], and the world market reach more than ten million tons in the recent year. Traditional enzymatic isomerization method suffers from reaction equilibrium and product inhibition resulting in a mixture of glucose and fructose and some byproducts, such as oligosaccharides and 5-hydroxymethylfurfural[39,40]. To obtain high-purity crystalline fructose necessary for pharmaceutical application or HFSC containing 90% of fructose (F90), a complicated and time-consuming separation process by aid of simulated moving bed (SMB) chromatography and excessive equipment investment are required. In this study, based on those substrate-specific mutants, we constructed a thermodynamically driven biosynthetic systems and successfully produced fructose from starch and sucrose with the excellent yield of 95% and highest proportion of fructose ($P_F$) of 98.8%. Especially, the commercial HFSC F90 can be easily obtained from sucrose through an easy-operated reaction without requirement of the complex SMB chromatography separation process. Such processes exhibit the advantages of noncofactor dependence, inexpensive carbon source, and thermodynamically favorable. Just like the commercial immobilized glucose isomerase[41,42], many materials and approaches[43–45] could be employed to co-immobilize four enzymes in our biosystem to increase the stability and repeatability in large-scale production. Therefore, our study presents an efficient and alternative approach for the industrially manufacturing fructose in the future.

Mannitol is a naturally occurring six-carbon sugar alcohol with wide applications in the food and pharmaceutical industry; approximately 150,000 tons is required in the market per year. At present, industrial-scale preparation of mannitol is divided into two processes, as follows: i) chemical isomerization of glucose to mannose or hydrolysis of sucrose to glucose and fructose; and ii) chemical hydrogenation of the mixture to mannitol and sorbitol[46,47]. For the chemical hydrogenation process, glucose and mannose are totally converted to sorbitol and mannitol, respectively, whereas fructose is converted into sorbitol and mannitol with the same ratio. The yield for mannitol from glucose remained 30–40% due to the reaction equilibrium between glucose and mannose[31]. When sucrose was the substrate, hydrogenation of the glucose–fructose mixture resulted in only 25% mannitol yield (Supplementary Fig. 14). In this study, we presented a thermodynamically favorable biosystem for manufacturing mannose from starch and sucrose with a yield of 94% and 45% (representing 90% of the theoretical value), respectively. If those two biosystems could be coupled with a technically

immature chemical hydrogenation reaction process in future, then the proportion of mannitol in the products will reach to 70–94%, which will be higher than that obtained by the traditional method. The bio-chemohybrid system will provide a highly efficient way for manufacturing mannitol.

Although application of G6P-specific mutant MG7 was not tested in this study, this mutant has high application potential in the production of glucose from low carbon compounds, such as one-carbon units or glycerol, through the construction of a cell-free chem-biosynthetic system[48], which has become the research hotspot in recent years. In summary, our study overcomes the thermodynamic equilibrium of traditional epi-/isomerization and provides an innovative and alternative strategy to high-yield manufacture bulk table sugars and high-value rare sugars and polyols without the requirement of complex separation processes and huge equipment investments.

## Methods

**Materials**. Strains and plasmids used in this study are listed in Supplementary Data 3. The chemicals including sucrose, glucose, fructose, mannose, maltodextrin, G1P, G6P, F6P, and M6P, AG6P, 2DG6P, R5P, and E4P, were purchased from Sigma-Aldrich (St Louis, MO, USA). The ampicillin for strain selection and iso-propyl-β-D−1-thiogalactopyranoside (IPTG) for inducing protein expression were obtained from Solarbio (Beijing, China). The maltodextrin with the dextrose equivalent of 4.0–7.0 were used as substrate for Pases screening system and in vitro production system. A Ni-NTA affinity chromatography column for protein purification was purchased from QIAGEN (Hilden, Germany).

**Enzyme mining, expression, purification and screening**. The enzyme Pase5 from *Thermotoga maritima* served as template to search candidate Pases from NCBI database. Sequences with different similarity were selected. Fifteen Pases were selected, and their protein sequences and accession codes were provided in Supplementary Data 1. To fit the protein expression in *E. coli*, genes of candidate Pases were codon-optimized and synthesized by Genscript (Nanjing, China) in expression plasmid pET21a. Then, the expression plasmids harboring Pases were transferred into *E. coli* BL21(DE3) for protein expression. One clone of the resulting recombinant strain *E. coli* was picked from LB plate and cultured in LB medium containing 100 mg/L ampicillin at 37 °C for overnight. And then, 1 mL seed culture was added into 50 mL LB medium containing 100 mg/L ampicillin and culture at 37 °C for 3-h to an optical density $OD_{600}$ of 0.6–0.8. The IPTG with a concentration of 0.5 mM was added into the medium to induce protein expression at 16 °C. After culture for 20 h, cells were collected and resuspended in 2 mL triethanolamine (TEA) buffer (50 mM, pH 6.5), and lysed by high-pressure homogenization at 4 °C. After centrifugation, the recombinant proteins Ts38HM6PP, Pase12, Pase14, StIA, TmGP, TkPGM, DtPGI/MPI, TtPGI, MF2, MM3, Tl4GT, TfPPGK, and BaSP were purified using Ni-NTA resin. In the screening system, the candidate Pases and mutants were obtained by heat treatment at 70 °C for 30 min. The enzymes medium with glycerol at a final concentration of 5% was stored at −20 °C for further use.

To discover phosphatases with specificity to particular substrates, the in vitro screening system was constructed. It was performed in a 200 μL reaction medium, which contained 10 g/L maltodextrin, 10 mM of PBS (pH 6.5), 5 mM of MgCl₂. 0.3 mg/mL of TmGP, 0.2 mg/mL of TkPGM, 0.2 mg/mL of DtPGI/MPI, and 60 μL of heat-treated or purified Pases, at 55 °C for 4 h. Then, the reaction was stopped by addition of 10% H₂SO₄ (0.5 μL). Samples were then analyzed by HPLC. The proportion of mannose/glucose/fructose in total monosaccharides ($P_{M/G/F}$) in products was calculated as followed:

$$P_M = \frac{\text{Mannose}(\text{g/L})}{(\text{Mannose}(\text{g/L}) + \text{fructose}(\text{g/L}) + \text{glucose}(\text{g/L}))} \times 100 \quad (1)$$

Given that some native phosphatases of *E. coli* displaying the catalytic acidity to substrates G6P, F6P and M6P, to avoid the interference of intracellular phosphatases on enzyme screening, the strain expressing control pET21a was meanwhile carried out under the same cultivation and expression conditions with that for candidate phosphatases. After heat treatment at 70 °C for 30 min, the crude enzyme activities of control strain to G1P, G6P, F6P, and M6P were measured. Our results showed that none of catalytic activity to those substrates were detected.

The concentration of glucose, fructose, and mannose were analysed using HPLC system (Agilent 1260) equipped with Sugar-Pak™ column (6.5 × 300 mm) with deionized water as a mobile phase. The standard curve for those chemicals were presented in Supplementary Fig. 15.

**Enzyme mutagenesis**. Site-directed mutations were conducted using PCR methods. The primers used for mutations were listed in Supplementary Data 2. Recombinant plasmid pET21a-Ts38HM6PP harbouring the gene of Pase1 from *Thermotoga sp*. 38H served as the template. The PCR products were digested by

DpnI at 37 °C for 2 h and followed by transformation into *E. coli* DH5α cells for plasmid construction. And then, the desired mutations were confirmed by PCR sequencing. The obtained plasmids containing genes of mutants were then transferred into *E. coli* BL21 (DE3) for protein expression. All the mutant enzymes were evaluated using the screening system. The beneficial M6P/G6P/F6P-sepecific single-site mutants served as templates to construct iterative mutants.

**Enzyme activity assay.** To measure the enzyme activity to substrates M6P, F6P, and G6P, the reaction mixtures (200 uL) containing 0.05 mg of Pases and mutants, 10 mM of TEA buffer (pH 6.5), 5 mM of MgCl$_2$, and 20 mM of substrates were used. The reactions were performed at 55 °C for 10–30 min and stopped by addition of 10% H$_2$SO$_4$ (0.5 μL). The formation of glucose, fructose, or mannose were measured by HPLC. One unit of enzyme activity was defined as the enzyme amount catalysing the formation of 1 μmol product per min. To quantitatively determine the specificity of one phosphatase I to substrate R1 over R2, the parameter ratio $S_{R1/R2}$ was calculated as followed:

$$S_{R1/R2} = \frac{\text{Enzyme activity of PaseI to substrate R1(U/mg)}}{\text{Enzyme activity of PaseI to substrate R2(U/mg)}} \quad (2)$$

We also introduced another parameter $RS_{R1/R2}^{Mutant/WT}$ to quantitatively describe the substrate preference switch from substrate R1 to substrate R2 compared with the WT as followed:

$$RS_{R1/R2}^{Mutant/WT} = \frac{S_{R1/R2} \ of \ mutant}{S_{R1/R2} \ of \ WT} \quad (3)$$

To measure the kinetic parameters of $K_m$ and $V_{max}$ of WT and mutants, the reaction mediums containing 10 mM of TEA buffer (pH 6.5), 5 mM of MgCl$_2$, various concentrations of M6P (2–60 mM), F6P (2–60 mM) or G6P (2–60 mM), and purified enzymes (0.05–0.5 mg) were constructed. The reactions were performed at 55 °C for 10–60 mins and stopped by adding 10% H$_2$SO$_4$ (0.5 μL). The products of mannose, fructose or glucose were quantitative measured by HPLC. The enzymatic activity was measured with the enzyme amount catalysing the formation of 1 μmol product per min. The kinetic parameters of $K_m$ and $V_{max}$ were obtained using the GraphPad Prism 5 software.

**Model preparation and B-factor calculations.** The cap-open model structure of Ts38HM6PP was developed by Phyre2 sever[49] using PDB 1nf2 as template, which shares 89% sequence identity with Ts38HM6PP. Then, the pose of substrate M6P in PDB 4zev and G6P in PDB 4zew, as well as the F6P geometry from PDB 3pt1 were superimposed into the model of Ts38HM6PP to obtain complex conformations. Each complex was refined by MD simulations as followed: i) 2000 steps of steepest descent followed by 1000 steps of conjugate gradient minimization was carried out to remove poor contacts in each system; ii) the system was slowly heated up to 300 K in 25,000 steps, and 50 ps density simulation and 500 ps constant pressure equilibration was performed to get well-settled pressure and temperature. iii) 10 ns unconstrained run was performed at 300 K and 1 atm with 2 fs integration time step. The trajectory of complex binding M6P was used to calculate the B-factor values of cap-closed conformation. The B-factor values of each residue in cap-open configuration was estimated by ResQ[50] using the cap-open model created by Phyre2 sever.

**Computational redesign and simulations.** The WT complexes binding particular three substrates served as the starting structure template for computational enzyme redesign. The amino acid residues around the sugar ring in cap domain were in silico mutated with 20 natural amino acid types using Rosetta Enzyme Design protocol. The substrate–enzyme binding energy for each mutant was compared with that of the wild type to prioritize mutations, and the top1 or 2 ranked mutants with lowest binding energy were selected for experimental characterization. The following design and minimization options were used: -enzdes -cst_design -design_min_cycles 3 -lig_packer_weight 1.8 -cst_min -chi_min -bb_min -cst_opt -packing:ex1 -packing:ex2 -packing:use_input_sc -packing:soft_rep_design. Geometrical constraints between the substrate and the desired interacting side chains were used to position substrates optimally for catalysis. Detail distance and angle constraints are listed in Supplementary Table 5.

The structure complex models of mutants were generated using Rosetta Enzyme Design application based on the WT-substrate complexes. The binding free energy in representative conformations were calculated using MM/GBSA method in AMBER16. The calculations were done on 100 frames equitably extracted from the 1 ns simulation trajectory. Protein–ligand interaction profiler[51] and Pymol was used to visualize models and construct graphical illustrative figures.

**In vitro cascade reaction.** The proof-concept synthesis of fructose from maltodextrins was conducted in reaction medium (1 mL) containing PBS buffer (10 mM, pH 6.5), IA-treated maltodextrin (10 g/L, 55 mM glucose equivalent), MgCl$_2$ (5 mM), TmGP (2 U/mL), TkPGM (1 U/mL), TtPGI (2 U/mL), and MF2 (2 U/mL). The reaction was performed at 55 °C for 12 h. To obtain IA-treated maltodextrin, the process was performed with substrate (maltodextrin)/enzyme (IA) ratio of 1500:1 at 80 °C for 2.5 h. To achieve high product yield, 1 U/mL of Tl4GT, 1 U/mL of TfPPGK and 5 mM of hexametaphosphate sodium were added to the reaction mixture at 6 h. When high concentration of maltodextrin (100 g/L, 555 mM glucose equivalent) was used, the reaction medium (20 mL) containing PBS buffer (30 mM, pH 6.5), 5 mM MgCl$_2$, TmGP (10 U/mL), TkPGM (5 U/mL), TtPGI (10 U/mL), and MF2 (10 U/mL) was carried out at 55 °C for 24 h. After reaction for 12 h, 5.0 U/mL of Tl4GT, 5.0 U/mL of TfPPGK, and 50 mM of hexametaphosphate were added to the reaction solution.

The synthesis of mannose from maltodextrins was conducted in reaction medium (1 mL) containing PBS buffer (10 mM, pH 6.5), IA-treated maltodextrin (10 g/L, 55 mM glucose equivalent), MgCl$_2$ (5 mM), TmGP (2 U/mL), TkPGM (1 U/mL), DtPGI/MPI (1 U/mL), and MM3 (2 U/mL). The reaction was performed at 55 °C for 12 h. Different Mg$^{2+}$ concentration of 0.5 mM, 1 mM, 2.5 mM and 5 mM were applied to test the effect of Mg$^{2+}$ concentration on substrate preference. The reaction conditions were also optimized by loading different DtPGI/MPI amount (0.5 U/mL, 1 U/mL, 2 U/mL, 4 U/mL and 8 U/mL). To completely conversion of maltodextrin, 1 U/mL of Tl4GT, 1 U/mL of TfPPGK and 5 mM of hexametaphosphate sodium were added to the reaction mixture after reaction for 6 h. Under 100 g/L of maltodextrin, the reaction medium (20 mL) containing 30 mM of PBS buffer, 1 mM of MgCl$_2$, 10 U/mL of TmGP, 5 U/mL of TkPGM, 20 U/mL of DtPGI/MPI, and 10 U/mL of MM3 was carried out at 55 °C for 24 h. After reaction for 12 h, 5.0 U/mL of Tl4GT, 5.0 U/mL of TfPPGK, and 50 mM of hexametaphosphate were added to the reaction solution.

To perform the proof-concept synthesis of fructose from sucrose, reaction medium (1 mL) containing PBS buffer (10 mM, pH 6.5), sucrose (10 g/L), MgCl$_2$ (5 mM), BaSP (1 U/mL), TkPGM (1 U/mL), TtPGI (2 U/mL), and MF2 (2 U/mL) was carried out at 55 °C for 12 h. The effect of BaSP amount (0.25 U/mL, 0.5 U/mL, 1 U/mL, and 2 U/mL) on production formation was measured. To test the fructose production under high concentration of sucrose (100 g/L), the reaction medium (20 mL) comprising of 20 mM of PBS buffer, 5 mM of MgCl$_2$, 2.5 U/mL of BaSP, 5 U/mL of TkPGM, 10 U/mL of TtPGI, and 10 U/mL of MF2 was carried out at 55 °C for 12 h.

For mannose production from sucrose, the reaction medium was altered to that contained MgCl$_2$ (1 mM), BaSP (0.5 U/mL), TkPGM (1 U/mL), DtPGI/MPI (4 U/mL), and MM3 (2 U/mL) was carried out at 55 °C for 12 h. When 100 g/L sucrose was used, the reaction medium (20 mL) comprising of 20 mM of PBS buffer, 1 mM of MgCl$_2$, 2.5 U/mL of BaSP, 5 U/mL of TkPGM, 20 U/mL of DtPGI/MPI, and 10 U/mL of MM3 was carried out at 55 °C for 12 h.

**Reporting summary.** Further information on research design is available in the Nature Research Reporting Summary linked to this article.

## Data availability

The data supporting the findings of this work are available within the paper and the Supplementary Information files. All relevant accession codes for the sequences and plasmids used in this study are available in Supplementary Data 1. The phosphatase protein sequences used in this work were downloaded from UniProt database (www.uniprot.org/) and NCBI database (www.ncbi.nlm.nih.gov/). Multi-sequence alignment of Ts38HM6PP with five phosphatases exhibiting catalytic ability to M6P, F6P and G6P using ClustalW program (www.genome.jp/tools-bin/clustalw) and the conserved motif is presented using ESPript 3.0 (https://espript.ibcp.fr/ESPript/cgi-bin/ESPript.cgi). The crystal structure of HAD phosphatase from *T. maritima* (PDB ID: 1NF2, www.rcsb.org/structure/1NF2) was used as a template for modeling of Ts38HM6PP. Source data are provided with this paper. Data is available from the corresponding authors upon request. Source data are provided with this paper.

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

## Acknowledgements

This work was supported by National Natural Science Foundation of China (No. 31972030 [JY]), National Key R&D Program of China (No. 2019YFA09004900 [JY]), Tianjin Synthetic Biotechnology Innovation Capacity Improvement Project (TSBICIP-KJGG-008-03 [JY] and TSBICIP-KJGG-003-06 [JY]), Chinese Academy of Sciences-Guangxi STS Project (KFJ-STS-QYZD-200 [YS]), Youth Promotion Association of Chinese Academy of Sciences (2021176 [JY]). We also appreciate Professor Chun You for kindly providing several pET20b derivative protein expression plasmids.

## Author contributions

C.T., J.Y., and C.L contributed equally to this work. They performed the enzyme mining, in silico design, and substrate specificity engineering experiments, analyzed the data, and prepared the manuscript. P. C. and T. Z constructed the synthetic system for manu-facturing hexoses. Y. Men. optimized the reaction conditions. J.Y., H.M., Y.S., and Y.Ma designed the experiment and revised the manuscript.

## Competing interests

The engineered mutants described in this paper are covered by patents CN202111513425.9, CN202111486922.4, and CN202111512164.9. C.T., J.Y., P.C., T.Z., Y.M., Y.S., and Y.M are listed as co-inventors of the patents. The rest of the authors declare no competing interests.
