## [Peer Review File · Nature Communications]

Engineering substrate specificity of HAD phosphatases and multienzyme systems development for the thermodynamic-driven manufacturing sugarsREVIEWER COMMENTS

Reviewer #1 (Remarks to the Author):

Y. Sun and Y.-H. Ma and their coworkers present a study in which structure-guided rational evolution of a phosphatase as the catalyst for the practical production of three different phosphorylated sugars is described: glucose-6-phosphate, fructose-6-phosphate and mannose-6-phosphate. This kind of enzyme promiscuity coupled with high selectivity has not been reported to date. In further impressive work, which builds on these results, four coordinated multienzyme systems were developed for converting sucrose and starch into fructose and mannose. The traditional problem of unfavorable reaction equilibrium of isomerization was solved. This is a biotechnological advance of considerable importance, a result of basic research carried out in a transdisciplinary manner.

The authors used haloacid dehalogenase (HAD)-like phosphatase. First, an effective screening system had to be developed, followed by enzyme mining, leading to the initial positive results. This was followed by rational enzyme design using structural data, including the identification of flexible regions as indicated by high B-factors. I wish to mention only a few of the highlights in the protein engineering campaign. In silico saturation mutagenesis at several hotspots was performed based on highly reduced amino acid alphabets (only a few specific amino acids presumably having the right steric properties). Based on the consensus concept, 7 other residues were mutated rationally. More than a dozen improved variants were identified and characterized in the laboratory. Some were then iteratively combined. Moreover, the origin of enhanced activity and specificity was revealed by MD computations.

The manuscript is written in a clear and scholarly manner (English can be improved at one or two places). The overall results are highly innovative and live up to the high quality of Nat. Commun. This reviewer recommends acceptance following minor revision: The strategy of the rational enzyme design used in this study seems to be similar to the recently published Focused Rational Iterative Site-specific Mutagenesis (FRISM). The authors should mention this and cite the respective papers: a) H. Xu, J. Am. Chem. Soc. 2019, 141, 7934-7945; b) D. Li, et al, Methods Enzymol. 2020, 643, 225-242.

Reviewer #2 (Remarks to the Author):

Naturally, HAD superfamily phosphatases presents broad substrate promiscuity. Engineering of phosphatases for dedicated substrate specificity indeed remains great challenge. This manuscript describes the structure-guided engineering of a phosphatase and presents three variants with tailor-made preference for three phosphorylated sugars while simultaneously enhancement in catalytic efficiency. The substrate specificity engineering approaches would help to engineer other phosphatases for strict substrate specificity. Based on those mutants, authors further presented four thermodynamically-driven cell-free synthetic systems and produced two widely-used sugars from inexpensive sucrose and starch with excellent yield and product proportion. It helps to overcome the reaction equilibrium of isomerization in manufacturing monosaccharides and would provide an innovative way to produce other sugars. Overall this is an very interesting and meaningful work. I recommend publication after that some minor issues as followed have been solved.

1. It seemed that Dr. Volker Sieber also has mentioned its work in engineering the substrate specificity of phosphatase to substrate F6P and G6P. Although he did not present good results for substrate preference switch or enhancement as this manuscript, I suggest adding some discussions for the previous work in this paper.

2. The authors mentioned enzyme mining of 15 phosphatases for high substrate specificity to F6P, M6P, and G6P. Please provide the SDS-PAGE for protein expression of those phosphatases.

3. For the screening system, 60 uL of heat-treated crude proteins were used. The E. coli also have many phosphatases in cell. The authors should mention whether the intracellular phosphatases still

have enzyme activity to G6P, F6P and M6P after heat-treat at 70 °C for 30 min.

4. For the multienzyme system for fructose and mannose production, the authors presented good conversion results under low concentration of maltodextrin and sucrose (10 g/L). However, under high concentration with 100 g/L, it seemed that the conversion rate was lower than that under 10 g/L. The probable reason? The author should give some discussions for those results.

5. For the in vitro cascade reaction, some information seemed misleading in the materials and methods part.

For fructose from 10 g/L maltodextrin, the author indicated that the reaction contained TmGP (2 U/mL), TkPGM (1 U/mL), TtPGI (2 U/mL), and MF2 (2 U/mL). Why TkPGM was used with 1 U/mL rather than 2 U/mL ????? For this reaction, 1 U/mL of TI4GT, 1 U/mL of TfPPGK and 5 mM of hexametaphosphate sodium were added to push the reaction to completely conversion. When 100 g/L maltodextrin was used, the enzyme amount of TmGP, TkPGM, TtPGI, and MF2 increased by 5-fold; however, the usage of TI4GT and TfPPGK increased by 10-fold. The reason ?? In addition, for mannose production from 100 g/L of maltodextrin, we can get the information that TI4GT and TfPPGK were added into the reaction medium; however, the author did not present the information in materials and methods part.

So, I think that the authors should present clear information for their experiment information especially for this part.

Other minor concerns:

1. "Typically, and high fructose corn syrup (HFCS)" the word "and" deleted.
2. The author should give line number for the manuscript. Some grammatical mistakes should be revised.
3. In supplementary Figure 9 and 14. GM7? or MG7?

Point-to-point response

Title: Substrate specificity engineering of HAD phosphatases and coordinated multienzyme systems development facilitating thermodynamic-driven manufacturing sugars

Overall Response: We are grateful for the reviewers' positive evaluations and their valuable suggestions. Here we prepared this point-to-point response and highlight changes in red.

Reviewer #1 (Remarks to the Author):

1) Y. Sun and Y.-H. Ma and their coworkers present a study in which structure-guided rational evolution of a phosphatase as the catalyst for the practical production of three different phosphorylated sugars is described: glucose-6-phosphate, fructose-6-phosphate and mannose-6-phosphate. This kind of enzyme promiscuity coupled with high selectivity has not been reported to date. In further impressive work, which builds on these results, four coordinated multienzyme systems were developed for converting sucrose and starch into fructose and mannose. The traditional problem of unfavorable reaction equilibrium of isomerization was solved. This is a biotechnological advance of considerable importance, a result of basic research carried out in a transdisciplinary manner.

The authors used haloacid dehalogenase (HAD)-like phosphatase. First, an effective screening system had to be developed, followed by enzyme mining, leading to the initial positive results. This was followed by rational enzyme design using structural data, including the identification of flexible regions as indicated by high B-factors. I wish to mention only a few of the highlights in the protein engineering campaign. In silico saturation mutagenesis at several hotspots was performed based on highly reduced amino acid alphabets (only a few specific amino acids presumably having the right steric properties). Based on the consensus concept, 7 other residues were mutated rationally. More than a dozen improved variants were identified and characterized in the laboratory. Some were then iteratively combined. Moreover, the origin of enhanced activity and specificity was revealed by MD computations.

Our response:

Thank very much for the reviewer's positive comments.

2) The manuscript is written in a clear and scholarly manner (English can be improved at one or two places). The overall results are highly innovative and live up to the high quality of Nat. Commun. This reviewer recommends acceptance following minor revision: The strategy of the rational

enzyme design used in this study seems to be similar to the recently published Focused Rational Iterative Site-specific Mutagenesis (FRISM). The authors should mention this and cite the respective papers: a) H. Xu, *J. Am. Chem. Soc.* 2019, 141, 7934-7945; b) D. Li, et al, *Methods Enzymol.* 2020, 643, 225-242.

Our response:

Thank very much for the reviewer's positive comments. We agree with the reviewer's comments very much for that the strategy of the rational enzyme design used in this study was similar with the recently published Focused Rational Iterative Site-specific Mutagenesis (FRISM). Therefore, we have cited the corresponding papers in the reference part according to the reviewers' suggestions. We also added some discussions in the discussion part as followed:

Line 354-356: "This rational design strategy was similar to the recently published Focused Rational Iterative Site-specific Mutagenesis (FRISM) method, which has been applied to engineer a lipase to access all possible stereoisomers of chiral esters."^{34,35}

Reference:

[34] Xu, J., Cen, Y., Singh, W., Fan, J., Wu, L., Lin, X., Zhou, J., Huang, M., Reetz, M. T., & Wu, Q. Stereodivergent protein engineering of a lipase to access all possible stereoisomers of chiral esters with two stereocenters. *J. Am. Chem. Soc.* **141** (19), 7934–7945 (2019).

[35] Li, D., Wu, Q., & Reetz, M. T. Focused rational iterative site-specific mutagenesis (FRISM). *Methods in enzymology.* **643**, 225-242 (2020)

In addition, according to the reviewer's suggestions, the description of whole manuscript has been checked and improved.

Reviewer #2 (Remarks to the Author):

1) Naturally, HAD superfamily phosphatases presents broad substrate promiscuity. Engineering of phosphatases for dedicated substrate specificity indeed remains great challenge. This manuscript describes the structure-guild engineering of a phosphatase and presents three variants with tailor-made preference for three phosphorylated sugars while simultaneously enhancement in catalytic efficiency. The substrate specificity engineering approaches would help to engineer other phosphatases for stick substrate specificity. Based on those mutants, authors further presented four thermodynamically-driven cell-free synthetic systems and produced two widely-used sugars from inexpensive sucrose and starch with excellent yield and product proportion. It helps to overcome the reaction equilibrium of isomerization in manufacturing monosaccharides and would provide an innovative way to produce other sugars. Overall this is an very interesting and meaningful work. I recommend publication after that some minor issues as followed have been solved.

Our response:

Thank very much for the reviewer's positive comments.

2) It seemed that Dr. Volker Sieber also has mentioned its work in engineering the substrate specificity of phosphatase to substrate F6P and G6P. Although he did not present good results for substrate preference switch or enhancement as this manuscript, I suggest adding some discussions for the previous work in this paper.

Our response:

Thank very much for the reviewer's suggestions. According to the reviewer's suggestions, we have added the discussions in the discussion part and cited the corresponding reference as followed:

Line 345-346: "Although the efforts have been made to increase the substrate preference of one HAD phosphatase; however, significant preference switch has not been achieved.³³"

[33] Sutiono, S. Enzyme engineering of a haloacid dehalogenase-like phosphatase from *Thermotoga neopolitana* for optimization of substrate specificity. Master Thesis, Lund University, Germany, 2016.

3) The authors mentioned enzyme mining of 15 phosphatases for high substrate specificity to F6P, M6P, and G6P. Please provide the SDS-PAGE for protein expression of those phosphatases.

Our response:

Thank very much for the reviewer's comments. According to reviewer's suggestions, we have provided the corresponding data of SDS-PAGE for protein expression of those phosphatases in supplementary figure. 8 in the Supporting information part.

4) For the screening system, 60 uL of heat-treated crude proteins were used. The *E. coli* also have many phosphatases in cell. The authors should mention whether the intracellular phosphatases still have enzyme activity to G6P, F6P and M6P after heat-treat at 70 °C for 30 min.

Our response:

Thank very much for the reviewer's comments. Indeed, *E. coli* also have many native phosphatases in cell. Those phosphatases have been confirmed with the catalytic ability to substrates G6P, F6P, and M6P. To avoid the interference of intracellular phosphatases on enzyme screening, we also measure the crude enzyme activity of control strain harbouring pET20b to G1P, G6P, F6P, and M6P after heat treatment at 70°C for 30 min. Our results showed that none of catalytic activity to those substrates were detected. To clear describe this part, we added some information in the manuscript as followed:

Line 434-439: “Given that some native phosphatases of *E. coli* displaying the catalytic activity to substrates G6P, F6P and M6P, to avoid the interference of intracellular phosphatases on enzyme screening, the strain expressing control pET20b was meanwhile carried out under the same cultivation and expression conditions with that for candidate phosphatases. After heat treatment at 70°C for 30 min, the crude enzyme activities of control strain to G1P, G6P, F6P, and M6P were measured. Our results showed that none of catalytic activity to those substrates were detected.”

5) For the multienzyme system for fructose and mannose production, the authors presented good conversion results under low concentration of maltodextrin and sucrose (10 g/L). However, under high concentration with 100 g/L, it seemed that the conversion rate was lower than that under 10 g/L. The probable reason? The author should give some discussions for those results.

Our response:

Thank very much for the reviewer’s comments. We also noted that the conversion rate results under high substrate concentration were lower than that under low substrate concentration. It was probably due to the Maillard reaction between sugars and enzymes under high reaction temperature of 55°C. Under high substrate concentration, high amounts of enzyme were used. The Maillard reaction would decrease the enzyme activity of used enzymes, therefore decreased the conversion yield. The probable methods to solve this problem were lower the reaction temperature or usage of enzyme immobilization methods. We also some discussions in the manuscript as followed:

Line 305-308: “Here, we noted that the conversion yields under high substrate concentration of 100 g/L maltodextrin and sucrose were lower than that under low substrate concentration, which was probably due to the Maillard reaction between sugars and enzymes under the reaction medium.^{29,30}”

6) For the in vitro cascade reaction, some information seemed misleading in the materials and methods part.

For fructose from 10 g/L maltodextrin, the author indicated that the reaction contained TmGP (2 U/mL), TkPGM (1 U/mL), TtPGI (2 U/mL), and MF2 (2 U/mL). Why TkPGM was used with 1 U/mL rather than 2 U/mL ???? For this reaction, 1 U/mL of Tl4GT, 1 U/mL of TfPPGK and 5 mM of hexametaphosphate sodium were added to push the reaction to completely conversion. When 100 g/L maltodextrin was used, the enzyme amount of TmGP, TkPGM, TtPGI, and MF2 increased by 5-fold; however, the usage of Tl4GT and TfPPGK increased by 10-fold. The reason ?? In addition, for mannose production from 100 g/L of maltodextrin, we can get the information that Tl4GT and

TfPPGK were added into the reaction medium; however, the author did not present the information in materials and methods part.

So, I think that the authors should present clear information for their experiment information especially for this part.

Our response:

Thank very much for the reviewer's comments. We are so sorry for the unclear information for this part. We have checked the corresponding part and give clear information in the revised version.

Line 501-505: "When high concentration of maltodextrin (100 g/L, 555 mM glucose equivalent) was used, the reaction medium (20 mL) containing PBS buffer (30 mM, pH 6.5), 5 mM MgCl₂, TmGP (10 U/mL), TkPGM (5 U/mL), TtPGI (10 U/mL), and MF2 (10 U/mL) was carried out at 55°C for 24 h. After reaction for 12h, 5.0 U/mL of Tl4GT, 5.0 U/mL of TfPPGK, and 50 mM of hexametaphosphate were added to the reaction solution."

Line 514-518: "Under 100 g/L of maltodextrin, the reaction medium (20 mL) containing 30 mM of PBS buffer, 1 mM of MgCl₂, 10 U/mL of TmGP, 5 U/mL of TkPGM, 20 U/mL of DtPGI/MPI, and 10 U/mL of MM3 was carried out at 55°C for 24 h. After reaction for 12 h, 5.0 U/mL of Tl4GT, 5.0 U/mL of TfPPGK, and 50 mM of hexametaphosphate were added to the reaction solution."

7) Other minor concerns:

1. "Typically, and high fructose corn syrup (HFCS)" the word "and" deleted.
2. The author should give line number for the manuscript. Some grammatical mistakes should be revised.
3. In supplementary Figure 9 and 14. GM7? or MG7?

Our response:

Thank very much for the reviewer's comments.

The word "and" have been deleted according to the reviewer's suggestions.

Line 365-366: "Typically, high fructose corn syrup (HFCS) is widely used as sucrose substitute in soft drinks³⁸,"

According to the reviewer's suggestions, we have added the line number for the manuscript, revised the grammatical mistakes through the whole manuscript, and highlight changes in red.

According to the reviewer's comments, the mutant name has been corrected with MG7 in supplementary Figure 10 and 15.

REVIEWERS' COMMENTS

Reviewer #2 (Remarks to the Author):

This already good paper has been much improved. The authors have responded to the reviewers' comments, and the paper should be accepted for publication.